# *Salmonella* Enteritidis ST11 Prosthetic Valve Endocarditis Complicated by a Paravalvular Abscess: Case Report and Literature Review

**Shiori Kitaya** [1,2,*,†], **Shintaro Katahira** [3], **Hiroaki Baba** [1], **Yoshikatsu Saiki** [3], **Yukio Katori** [2], **Koichi Tokuda** [1] and **Hajime Kanamori** [1,*,†]

1   Department of Infectious Diseases, Internal Medicine, Tohoku University Graduate School of Medicine, Sendai 980-8575, Japan; hbaba48@med.tohoku.ac.jp (H.B.); tokuda@med.tohoku.ac.jp (K.T.)
2   Department of Otolaryngology, Head and Neck Surgery, Tohoku University Graduate School of Medicine, Sendai 980-8575, Japan; yukio.katori.d1@tohoku.ac.jp
3   Division of Cardiovascular Surgery, Tohoku University Graduate School of Medicine, Sendai 980-8575, Japan; shinkatahira@med.tohoku.ac.jp (S.K.); yoshisaiki@med.tohoku.ac.jp (Y.S.)
*   Correspondence: shiori.kitaya.b7@tohoku.ac.jp (S.K.); kanamori@med.tohoku.ac.jp (H.K.)
†   These authors contributed equally to this work.

**Abstract:** *Salmonella* infection typically causes self-limiting gastroenteritis. However, in rare cases, it can lead to prosthetic valve endocarditis (PVE), especially in older adults with a history of valve replacement surgery. In this case study, we describe a case of *Salmonella* PVE in a man with a prosthetic aortic valve. Complications of PVE include abscess formation, which is associated with increased mortality. If a patient with a history of prosthetic valve replacement presents with symptoms suggestive of gastroenteritis or bacteremia, a thorough investigation should be conducted with suspicion of PVE. The prognosis of *Salmonella* PVE can be improved by promptly initiating appropriate antibiotics and administering them for an adequate duration, as well as by considering surgical intervention when necessary. Additionally, confirming negative blood cultures after treatment of *Salmonella* bacteremia is important to prevent the development of PVE and paravalvular abscesses.

**Keywords:** *Salmonella* Enteritidis ST11; prosthetic valve endocarditis; paravalvular abscess

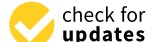



## 1. Introduction

*Salmonella* is a common cause of enteritis and accounts for approximately 5–10% of cases of bacteremia [1,2]. It is also a rare cause of destructive endocarditis, including prosthetic valve endocarditis (PVE), which occurs in only 0.2–0.4% of cases of salmonellosis [3,4]. PVE has a high mortality rate due to complications such as conduction abnormalities, septic embolism, perforation, valve rupture, and paravalvular abscess (PA) [3,5]. Risk factors for *Salmonella* endocarditis include congenital heart disease, immunodeficiency (including human immunodeficiency virus (HIV) infection), diabetes mellitus, and a history of heart valve replacement [3,6]. A limited number of reports of PVE caused by *Salmonella* have been published. Herein, we report the case of a patient with a history of valve replacement who developed bacteremia from *Salmonella* gastroenteritis following the consumption of raw eggs and subsequently developed PVE and PA. Additionally, we review the literature on *Salmonella* PVE to describe its microbiological and clinical characteristics.

## 2. Case Report

A 66-year-old man with a prosthetic aortic valve developed *Salmonella* PVE and was admitted to our hospital. He had undergone catheter ablation for paroxysmal atrial fibrillation 10 years previously and aortic valve replacement for severe aortic regurgitation 3 years previously. He also had a history of hypertension and chronic heart failure. Prior to admission, he had been treated for *Salmonella* gastroenteritis and bacteremia at another

hospital after eating raw eggs and was treated with intravenous piperacillin–tazobactam (4.5 g every 8 h) and oral levofloxacin (500 mg daily) for 1 week each.

On admission, he had a temperature of 37.5 °C, blood pressure of 103/73 mmHg, pulse of 78 beats/min, and 98% oxygen saturation in room air (Figure 1). His chest was clear on auscultation. Cardiovascular examination revealed a regular rate and rhythm, with normal first and second heart sounds. Blood tests revealed normal kidney and liver function, a white blood cell count of 8100/μL, and C-reactive protein level of 5.0 mg/L. An HIV antibody test was negative.

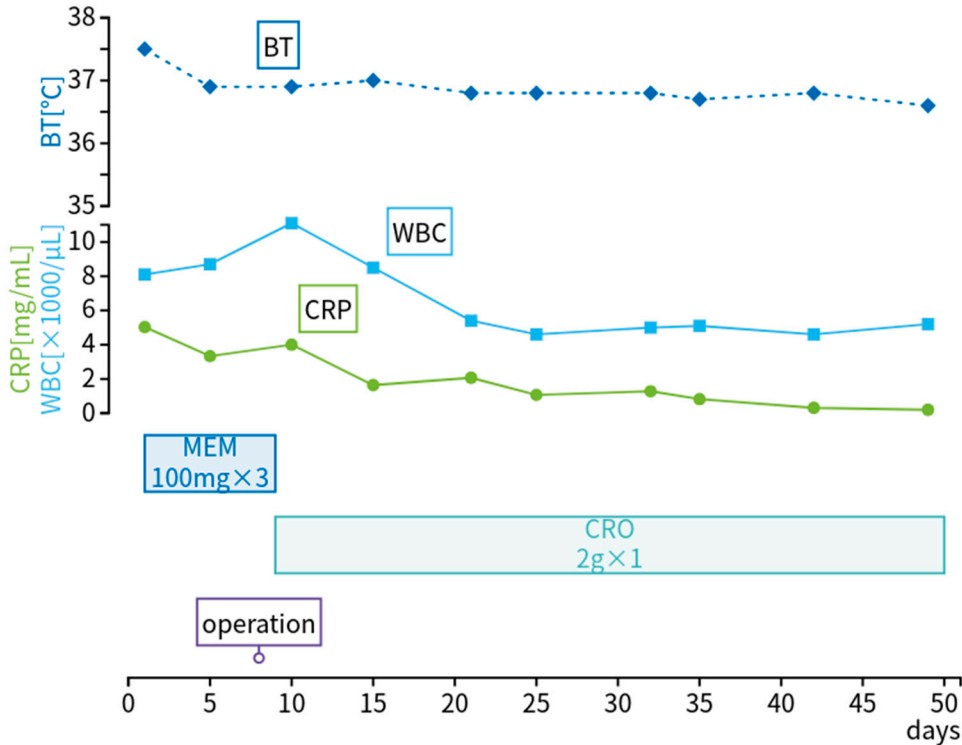

**Figure 1.** Vital signs and clinical course of the disease and treatment during hospitalization. BT, body temperature; CRP, C-reactive protein; CRO, ceftriaxone; MEM, meropenem; WBC, white blood cell.

Transthoracic echocardiography revealed free space at the aortic annulus and aortic paravalvular regurgitation (Figure 2a). Chest computed tomography (CT) revealed erosive marginal irregularities around the prosthetic valve (Figure 2b). Whole-body CT revealed no signs of embolism. The patient met one major (echocardiogram with abscess) and three minor (predisposing heart condition of prior prosthetic valve implantation, history of high fever, and serological evidence of active infection consistent with endocarditis) modified Duke diagnostic criteria for infective endocarditis [7]. He also met the criteria for complicated endocarditis, which include heart failure, intracardiac abscess or fistula, native valve *Staphylococcus aureus* endocarditis, or systemic embolization [8]. He was diagnosed with *Salmonella* PA and underwent surgical valve replacement and debridement 8 days after admission (Figure 2c).

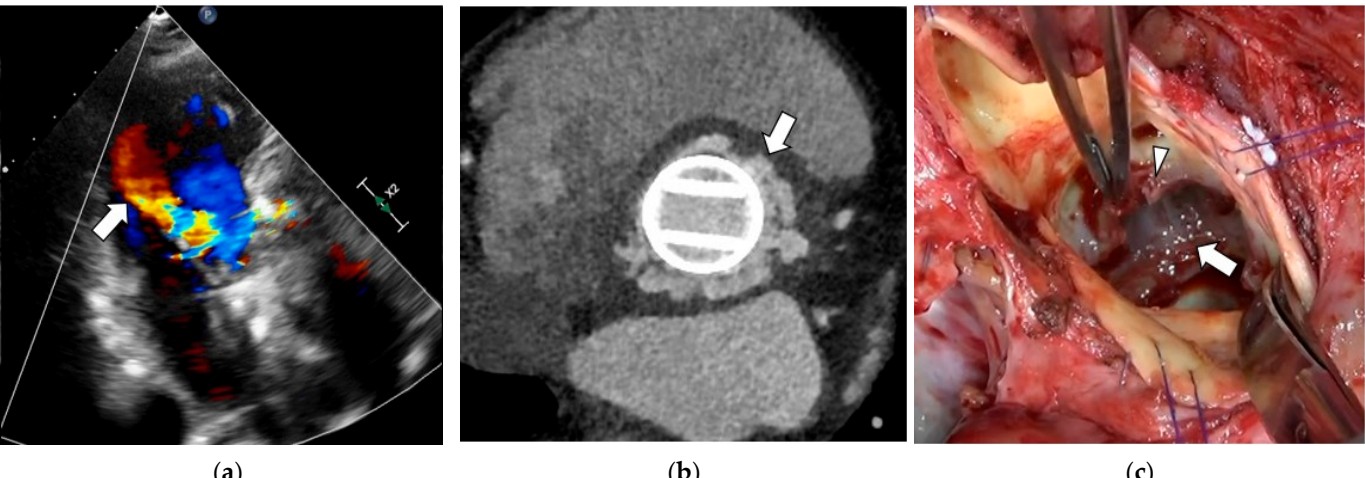

**Figure 2.** Transthoracic echocardiography, computed tomography, and surgical findings in *Salmonella* prosthetic valve endocarditis complicated by a paravalvular abscess. (**a**) Transthoracic echocardiography shows the presence of a free space at the aortic annulus and new aortic paravalvular regurgitation (arrow). (**b**) Chest computed tomography reveals erosive margin irregularities in the tissue enveloping the aortic prosthetic valve (arrow). (**c**) The suture at the aortic valve annulus is nearly detached, and a purulent cavity has formed around the aortic valve annulus (arrow). Granulation tissue is present around the valve ring tissue (arrowhead). It was thoroughly cleaned and replaced with a new prosthetic valve.

The granulation tissue collected intraoperatively was sent for whole-genome sequencing and serotyping. The *Salmonella* strains underwent slide agglutination testing using O-antisera (Denka Seiken Co., Tokyo, Japan) to determine their O-group antigen. Additionally, tube agglutination with H-antisera (Denka Seiken, Tokyo, Japan) was performed to identify the flagellar antigens. Serotyping was conducted based on the O-group and H-group antigens according to the Kauffman–White scheme [9]. DNA extraction was performed with a QIA amp DNA Mini Kit (Qiagen, Hilden, Germany). Library preparation and DNA fragmentation were performed on genomic DNA extracted from the sample using a Nextera DNA Flex Library Prep Kit (Illumina, San Diego, CA, USA), and Nextera DNA CD Index was used as the index adapter, according to the manufacturer's instructions. Whole-genome sequencing was performed using Illumina iSeq 100 with paired-end 150 bp reads. This identified the causative agent as *Salmonella enterica* subspecies *enterica* serovar Enteritidis (*S.* Enteritidis) sequence type (ST)11, which harbored the virulence genes *spvB*, *spvC*, and *spvR* (NCBI BioProject PRJNA015399).

Intravenous meropenem (MEM) (1 g every 8 h for 1 week) was initiated and subsequently de-escalated to intravenous ceftriaxone (CRO) (2 g daily for 6 weeks) based on antimicrobial susceptibility results and improvement in the patient's clinical condition. His postoperative recovery was uneventful, and he was discharged 51 days after admission. To prevent recurrence, he was prescribed prophylactic oral trimethoprim–sulfamethoxazole (SXT) (800/160 mg every 12 h) and advised against consuming raw eggs.

## 3. Literature Search and Selection Criteria

### 3.1. Literature Search

A literature search for *Salmonella* PVE was conducted from May 1965 until August 2022 using the PubMed database. The search was performed using the following keywords: "*Salmonella*", "prosthetic valve endocarditis (PVE)", and "paravalvular abscess (PA)". This study was approved by the Human Ethical and Clinical Trial Committee of Tohoku University Hospital (approval: 2019-1-270).

*3.2. Selection Criteria for Case Reports*

Case reports were included in the analysis if they met the following criteria: (1) the diagnosis of *Salmonella* PVE was confirmed, and (2) data for demographic and clinical characteristics were reported. S.K. (Shiori Kitaya) screened each title and abstract for initial eligibility based on the inclusion criteria, then performed a full text review to determine final study eligibility.

*3.3. Data Extraction*

The following variables were extracted: the author, year of publication, patient characteristics (e.g., age, sex); organism; preexisting heart disease and affected sites; types and positions of preexisting prosthetic valves; location of endocarditis; presence or absence of gastroenteritis and bacteremia; treatment (presence or absence of surgical treatment, chronic suppressive therapy, and duration of antibiotic therapy); presence or absence of disease relapse; complications, and outcome.

*3.4. Statistical Analysis*

We used the Fisher's exact test to compare the proportions of categorical variables between the two groups. To compare the incidence of PVE between the aortic, mitral, and tricuspid valves, we used Fisher's exact test and corrected the *p*-values using the Ryan–Holm step-down Bonferroni procedure. The analysis was performed using the JMP Pro 16 statistical analysis software (SAS Institute, Cary, NC, USA, 2021). Differences were considered significant if the *p*-value was <0.05.

**4. Results**

The literature review revealed 21 case reports on *Salmonella* PVE and one case report on *Salmonella* native valve endocarditis [3,4,10–28]. The results of the literature review and our case are summarized in Table 1 and described in more detail in Table S1. *Salmonella* PVE occurred predominantly in middle-aged and older adults (median age: 59 years), with no significant differences observed between sexes. Mechanical valves were used in a higher percentage of cases (11 cases, 50%) than prosthetic valves (6 cases, 27%), although this difference was not statistically significant (4 cases where the valve type is unknown). The review also identified two cases of *Salmonella* native valve endocarditis [20,29]. (One case lacked detailed information and was excluded from the review.) Additionally, two cases involved both prosthetic and native valves [14,19], and one case involved only native valve [3].

Among the cases reviewed in this study, only nine patients (41%) reported symptoms of gastroenteritis. The predominant causative organism was S. Enteritidis (eleven cases (50%)), followed by S. Typhimurium (four cases (18%)).

Regarding treatment, six cases (27%) were treated with antimicrobial therapy alone, without surgical intervention, and sixteen cases (73%) underwent surgical intervention in addition to antimicrobial therapy, including procedures such as valve replacement or debridement. The case fatality rate was higher in patients who did not receive surgical intervention (two cases out of six, 33%) than in those who received surgical intervention (two cases out of sixteen, 13%).

The most commonly used antibiotics were CRO (ten cases, 45%) and ciprofloxacin (CIP) (ten cases, 45%), followed by gentamicin (GEN) (eight cases, 36%). CIP was the most widely used agent for chronic oral antimicrobial suppression (eight cases out of eleven, 73%), followed by SXT (two cases out of eleven, 18.2%) (Table S1).

Complications occurred in approximately half of the cases, with abscesses being the most common (seven cases, 32%), followed by conduction disturbances (four cases, 18%), and emboli (three cases, 14%). The major sites of abscess formation were the around the aortic valve or root.

**Table 1.** Characteristics of *Salmonella* Prosthetic Valve Endocarditis.

| First Author, Year | Age, Sex | Organisms | Preexisting Prosthetic Valves (Location of Valves) | Location of Endocarditis | Gastro-Enteritis | Abscess (Location) | Other Complications | Treatment | Outcome |
|---|---|---|---|---|---|---|---|---|---|
| Weinstein [10], 1965 | 44, M | *Salmonella* Schwarzengrund | Hufnagel (Ao) | Ao | N/A | Yes (N/A) | Yes (cerebral embolism) | Antibiotics alone (CIP) | Died |
| Fraser [11], 1967 | 56, F | *Salmonella* Hirschfeldii, *Salmonella* Choleraesuis | Bahnson (Ao) | Ao | No | No | No | Surgery and antibiotics (AMP, CHL, MET, and STR) | Died |
| Yamamoto [12], 1974 | 50, F | *Salmonella* Enteritidis | N/A (Mit) | Mit | N/A | No | N/A | Surgery and antibiotics (AMP and TET) | Survived |
| Shanson [13], 1977 | 52, M | *Salmonella* Enteritidis | Starr–Edwards (Ao) | Ao | N/A | No | N/A | Surgery and antibiotics (AMP, MET, CFZ, GEN, CHL, and SXT) | Died |
| Choo [14], 1992 | 62, F | *Salmonella* Heidelberg | Hancock (Mit, Tri) | Ao, Mit, Tri | Yes | No | Yes (Tri regurgitation, AF, left-axis deviation, and incomplete RBBB) | Surgery and antibiotics (AMP, CRO, GEN, and VAN) | Survived |
| Hufnagel [15], 1993 | 65, F | *Salmonella* Enteritidis | N/A (Mit) | Mit | No | N/A | N/A | Surgery and antibiotics (CIP) | Survived |
| Lee [16], 1994 | 42, M | Group B | Björk–Shiley (Ao, Mit) | Mit | N/A | No | Yes (AF and RBBB) | Antibiotics alone (CRO) | Died |
| Fukushima [17], 1996 | 58, M | *Salmonella* Typhimurium | N/A (Ao, Mit, Tri) | Ao, Mit | No | No | No | Surgery and antibiotics (IPM) | Survived |
| Miyamoto [18], 1997 | 59, M | *Salmonella* Enteritidis | Carpentier–Edwards (Ao) | Ao | Yes | Yes (paravalvular) | Yes (4:3 Wenckebach block) | Surgery and antibiotics (CIP, CRO, GEN, MTZ, and VAN) | Survived |
| Goerre [19], 1998 | 79, M | *Salmonella* Enteritidis | Carpentier–Edwards (Ao) | Ao, Mit | Yes | No | No | Antibiotics alone (AMX, CIP, CVA, and NET) | Survived |
| Pliakos [20], 1998 | 65, M | N/A | None | Ao | No | Yes (aortic ring) | Yes (disruption of the aortic valve and aortic root, perforation of the noncoronary cusp, and fistula) | Surgery and antibiotics (CHL and CIP) | Survived |
| Gunalingam [21], 2000 | 69, M | *Salmonella* Typhimurium | N/A (Ao) | Ao | Yes | No | Yes (peripheral embolism) | Surgery and antibiotics (AMP and GEN) | Survived |

**Table 1.** *Cont.*

| First Author, Year | Age, Sex | Organisms | Preexisting Prosthetic Valves (Location of Valves) | Location of Endocarditis | Gastro-Enteritis | Abscess (Location) | Other Complications | Treatment | Outcome |
|---|---|---|---|---|---|---|---|---|---|
| Urfer [22], 2000 | 80, F | *Salmonella* Braenderup | N/A (Mit) | Mit | N/A | No | No | Antibiotics alone (CIP) | Survived |
| Keller [23], 2001 | 85, F | *Salmonella* Typhimurium | Björk–Shiley (Ao) | Ao | Yes | No | Yes (AF) | Surgery and antibiotics (AMX, CIP, and GEN) | Survived |
| Aribas [24], 2002 | 62, F | *Salmonella* Enteritidis | N/A (Mit) | Mit | Yes | No | No | Surgery and antibiotics (CRO, NET, and VAN) | Survived |
| Gönen [25], 2004 | 51, M | *Salmonella* Enteritidis | Kay–Shiley (Mit) and Björk–Shiley (Ao) | Mit | Yes | No | No | Surgery and antibiotics (SAM, CIP, GEN, and PEN) | Survived |
| Mutlu [3], 2009 | 69, F | *Salmonella* Enteritidis | N/A (Ao) | Mit, Tri | No | No | No | Surgery and antibiotics (CRO and LVX) | Survived |
| Clohessy [26], 2012 | 75, F | *Salmonella* Typhimurium | N/A (Ao) | Ao | No | Yes (aortic root) | Yes (pacemaker infection) | Surgery and antibiotics (AZM, CIP, and CRO) | Survived |
| Lam [27], 2017 | 55, M | *Salmonella* Enteritidis | N/A (Ao, Mit) | Ao, Mit | No | Yes (aortic root) | Yes (aortic root graft and pacemaker lead infections) | Antibiotics alone (CRO) | Survived |
| Connolly [28], 2021 | 54, F | *Salmonella* Enteritidis | St Jude (Mit) and Carbomedics Top Hat (Ao) | Mit | Yes | No | No | Antibiotics alone (AMX, CIP, CRO, GEN, MEM, MTZ, RIF, and VAN) | Survived |
| Alhamadh [4], 2022 | 56, F | Group C and D | N/A (Ao, Mit) | Mit | No | Yes (aortic root) | Yes (splenic infarction due to septic embolism, aortic pseudoaneurysm, and kidney injury) | Surgery and antibiotics (CRO and GEN) | Survived |
| Present case, 2022 | 66, M | *Salmonella* Enteritidis | ON-X (Ao) | Ao | Yes | Yes (paravalvular) | No | Surgery and antibiotics (CRO, LVX, MEM, and TZP) | Survived |

Abbreviations: AF, atrial fibrillation; AMP, ampicillin; AMX, amoxicillin; Ao, aortic; AZM, azithromycin; CFZ, cefazolin; CHL, chloramphenicol; CIP, ciprofloxacin; CRO, ceftriaxone; CVA, clavulanate; F, female; GEN, gentamicin; IPM, imipenem; LVX, levofloxacin; M, male; MEM, meropenem; MET, methicillin; Mit, mitral; MTZ, metronidazole; N/A, Not available (Details not provided, unknown.); NET, netilmicin; OFX, ofloxacin; PEN, penicillin; PVE, prosthetic valve endocarditis; RBBB, right bundle branch block; RIF, rifampin; SAM, ampicillin-sulbactam; STR, streptomycin; SXT, trimethoprim-sulfamethoxazole; TET, tetracycline; Tri, tricuspid; TZP, piperacillin-tazobactam; VAN, vancomycin.

## 5. Discussions

### 5.1. Epidemiological Characteristics

*Salmonella* endovascular infections occur mostly in individuals aged ≥50 years with underlying cardiovascular disease [30,31], and 25% of patients with bacteremia aged ≥50 years develop endocarditis or mycotic aneurysms [2,32], consistent with our literature review. Generally, endocarditis is more common in patients with prosthetic heart valves; however, *Salmonella* can cause native valve endocarditis in patients without prosthetic valves with or without a history of cardiac disease [20,29]. In our literature review, the prosthetic valve affected by PVE was the aortic valve in 11 cases (50%), the mitral valve in 11 cases (50%), and the tricuspid valve in one case (5%) ($p < 0.001$). Endocarditis occurred primarily in prosthetic valves, but in some cases, vegetations formed on both prosthetic and native valves [14,19] or only on native valves [3]. Therefore, in patients with suspected endocarditis, both prosthetic and native valves should be examined.

Over 95% of *Salmonella* infections are acquired from food [32,33]. Although gastroenteritis is a common symptom, it is often absent in cases of *Salmonella* endocarditis. Only 30% of patients with *Salmonella* endocarditis are reported to experience accompanying diarrhea [34]. Of the 22 cases reviewed, only nine patients (41%) had accompanying gastroenteritis.

In this review, the predominant causative pathogen was *S.* Enteritidis (11 cases, 50%), followed by *Salmonella* Typhimurium (four cases, 18%). Previous reviews have also reported a predominance of *S.* Enteritidis [14,17]. *S.* Enteritidis infection has the following three noteworthy epidemiological characteristics: (1) host specificity, (2) egg contamination, and (3) resistance to adverse conditions. *S.* Enteritidis exhibits a particular affinity for the reproductive organs of laying hens and can persist in this environment [35]. In contrast to other serotypes that primarily cause contamination on the outer surface of eggs, *S.* Enteritidis can penetrate the eggshell and proliferate in the egg white, which contains antibacterial factors that inhibit bacterial growth [36]. Owing to these characteristics, consumption of contaminated eggs and egg products is a major global source of *S.* Enteritidis infection and a cause of human salmonellosis. ST11, as detected in this case, is a clinically significant ST because it can persist in different host and environmental sample types [37] and has caused outbreaks, primarily in Europe [38]. In this review, several patients with PVE caused by *S.* Enteritidis did not report gastrointestinal symptoms. In most of these cases, the route of infection was unclear.

Generally, Gram-negative bacteria rarely cause PVE because they lack adherence factors to attach to artificial heart valves [23]. However, *Salmonella* exhibits increased stickiness owing to fibronectin peptide [39] and can cause intravascular infection by infecting atherosclerotic plaques or arterial aneurysms [40]. Previous reports have stated that the type of prosthetic valve does not affect the development of *Salmonella* PVE [41]. In this review, mechanical valves were more commonly infected than bioprosthetic valves (11/17 cases, 65% vs. 6/17 cases, 35%), although this difference was not significant.

### 5.2. Treatment

In this review, 73% (16/22) of patients required surgery, such as valve replacement or debridement, in addition to antibiotics, and only 27% (6/22) of patients were treated with antimicrobial therapy alone. In *Salmonella* endocarditis, early diagnosis and surgical intervention are essential to reducing the mortality rate and incidence of complications such as conduction disturbances, septic emboli, valve perforation or rupture, and PA [42]. Among the 16 patients in this review who underwent surgery, only two fatalities (13%) and eight cases (50%) of complications occurred, including conduction disturbances, septic emboli, and PA. In contrast, among the six patients who did not undergo surgery, two fatalities (33%) and three cases (50%) of complications occurred, although the difference in the incidence of complications and mortality rates in patients treated surgically and those treated with antibiotics alone was not statistically significant. This may support the importance of surgical intervention to improve the prognosis of *Salmonella* PVE.

Third-generation cephalosporins and fluoroquinolones are currently the first-line therapy for *Salmonella* endocarditis [25]. However, certain *Salmonella* strains are resistant to CRO [43,44]. There is a report stating that combination treatment with a third-generation cephalosporin and a fluoroquinolone should be used for life-threatening infections while susceptibility results are pending [2]. In this review, CRO (10/22 cases, 45%), CIP (10/22 cases, 45%), and GEN (8/22 cases, 36%) were the most frequently used antibiotics. Additionally, in this case, we de-escalated from MEM to CRO based on the antibiotic susceptibility results, the patient's relatively good general condition, and the risk of side effects, such as convulsions and QT prolongation, with long-term CIP use.

Antimicrobial treatment should be continued for at least 6 weeks when surgical intervention is performed but can be extended to years to provide chronic oral antimicrobial suppression in patients who are not candidates for surgery or if there is a possibility of residual bacteremia (such as when prosthetic material cannot be removed or when another infectious focus, such as osteomyelitis, is present) [2]. Considering the risk of infection due to the implanted aortic prosthetic valve, in this case, the patient was treated with STX following 6 weeks of treatment with intravenous CRO for chronic antimicrobial suppression. STX was chosen based on its (1) excellent oral absorption, (2) relatively narrow spectrum of activity, and (3) low cost. However, in the review, CIP was the most widely used agent for chronic antimicrobial suppression (8/11 cases, 73%). Moreover, side effects such as rashes, gastrointestinal disorders, and hematologic disorders have been reported with STX [30]. Therefore, patients' overall condition must be monitored, and options for oral antimicrobial therapy must be carefully considered.

*Salmonella* can lead to a chronic carrier state [45]. In this case, the patient relapsed after a 2-week course of antibiotic treatment at the first hospital, necessitating admission to our hospital. Negative blood culture tests were not confirmed at discharge from the first hospital; therefore, the patient might have developed persistent bacteremia or a chronic carrier state during this period. Similar cases of recurrence have been previously reported [14,17,19]. Therefore, negative blood culture results must be confirmed after treatment for *Salmonella* bacteremia to prevent the development of persistent bacteremia or a chronic carrier state.

### 5.3. Complications

Complications of *Salmonella* PVE include conduction abnormalities, septic embolism, perforation or rupture, and PA [3,5]. Complications occurred in approximately half of the cases reviewed, most commonly abscesses (7/22 cases, 32%), conduction abnormalities (4/22 cases, 18%), embolism (3/22 cases, 14%), and infections in other sites (2/22 cases, 9%). The predominant site of abscess formation was around the aortic valve or root. Although the prevalence of aortic and mitral valves as PVE sites did not differ significantly, the results suggest a higher incidence of abscesses around the aortic valve or root.

### 6. Conclusions

In summary, a history of prosthetic valve replacement is a risk factor for *Salmonella* endocarditis, and when complications such as abscesses occur, the symptoms can become severe, leading to increased mortality rates. Gastrointestinal symptoms are often absent in *Salmonella* PVE. As *Salmonella* can also cause native valve endocarditis, both prosthetic and native valves have to be assessed. If a patient who has undergone prosthetic valve replacement exhibits symptoms indicating gastroenteritis or bacteremia, a thorough examination should be performed, including the use of ultrasound and other diagnostic measures, keeping in mind the possibility of PVE. In cases of *Salmonella* PVE, appropriate antibiotics must be promptly administered for an adequate duration, and surgical intervention must be considered when necessary, as these measures are essential to improving the prognosis. Additionally, clinicians should confirm that blood cultures are negative after treatment of *Salmonella* bacteremia to prevent the development of PVE and PA.

**Supplementary Materials:** The following supporting information can be downloaded at: https://www.mdpi.com/article/10.3390/applmicrobiol3030073/s1, Table S1: Characteristics of *Salmonella* Prosthetic Valve Endocarditis.

**Author Contributions:** Conceptualization, S.K. (Shiori Kitaya) and H.K.; methodology, S.K. (Shiori Kitaya) and H.K.; investigation, H.K.; data curation, S.K. (Shiori Kitaya); writing—original draft preparation, S.K. (Shiori Kitaya); writing—review and editing, S.K. (Shiori Kitaya), S.K. (Shintaro Katahira), H.B., Y.S., Y.K., K.T. and H.K. All authors have read and agreed to the published version of the manuscript.

**Funding:** This study was partially supported by joint research between Tohoku University and NBC Meshtec Inc, 2-50-3 TOYODA, HINO, TOKYO 191-0053, JAPAN.

**Institutional Review Board Statement:** The study was conducted according to the guidelines of the Declaration of Helsinki and approved by the Ethics Committee of Tohoku University Graduate School of Medicine (approval: 2019-1-270).

**Informed Consent Statement:** Written informed consent was obtained from the patient for the publication of this case report.

**Data Availability Statement:** The data underlying the case report cannot be shared publicly to protect the privacy of the patient. All the relevant data regarding the literature review are included in this report.

**Acknowledgments:** We would like to thank Yumiko Takei from the Department of Infectious Diseases, Internal Medicine, Tohoku University Graduate School of Medicine, Sendai, Japan for her technical assistance with the analysis of bacterial isolates.

**Conflicts of Interest:** The authors declare no conflict of interest.

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
