# Peer review of "Salmonella Enteritidis ST11 Prosthetic Valve Endocarditis Complicated by a Paravalvular Abscess: Case Report and Literature Review"

_2673-8007, doi:10.3390/applmicrobiol3030073_

Round 1

Reviewer 1 Report

The overall case presentation and literature search and analyses are interesting.  I have no major issues with the content.  However, I think a minor restructure is required.  Presently there is no "Results" section and analyses are presented in Discussions.  I would suggest having a Results section.

No comments

Author Response

Reviewer 1

The overall case presentation and literature search and analyses are interesting. I have no major issues with the content. However, I think a minor restructure is required. Presently there is no "Results" section and analyses are presented in Discussions. I would suggest having a Results section.

Response: Thank you for reviewing our manuscript and for your positive feedback and constructive comment. We have done as you recommended and restructured the manuscript to include a separate Results section in which we report the findings of the literature review, as you suggested.

Reviewer 2 Report

The introduction and case report is fine. It contributes to an increased knowledge in this rare type of PVE.

However the presentation of the literature review could be improved. It is difficult to find out if the review describes the results in Table 1 or from other authors. Maybe the discussion part  could be replaces by Results of literature review and followed by Discussion.

In Table 1 there are abbreviations for antibiotics  I am not familiar with, ASM has a proper list. MEPM for example is not explained. 

See above. In Table 1 there are abbreviations for antibiotics  I am not familiar with, ASM has a list. MEPM is not explained. 

Author Response

Reviewer 2

The introduction and case report is fine. It contributes to an increased knowledge in this rare type of PVE. However the presentation of the literature review could be improved. It is difficult to find out if the review describes the results in Table 1 or from other authors. Maybe the discussion part could be replaced by Results of literature review and followed by Discussion.

Response: We thank the reviewer for this comment. We have added a Results section in which we report the findings of the literature review, as you suggested. Additionally, we have made some modifications to the discussion content to enhance its clarity.

In Table 1 there are abbreviations for antibiotics I am not familiar with, ASM has a proper list. MEPM for example is not explained.

Response: Thank you for suggesting that we use the ASM list of antibiotic abbreviations. As you suggested, we have replaced the antibiotic abbreviations with the ASM antibiotic abbreviations in the main text and Figure 1.

Round 2

Reviewer 1 Report

No further comments